# Genome-Wide DNA Methylation Patterns of Muscle and Tail-Fat in DairyMeade Sheep and Mongolian Sheep

**DOI:** 10.3390/ani12111399

**Published:** 2022-05-29

**Authors:** Rongsong Luo, Xuelei Dai, Li Zhang, Guangpeng Li, Zhong Zheng

**Affiliations:** 1State Key Laboratory of Reproductive Regulation & Breeding of Grassland Livestock, School of Life Sciences, Inner Mongolia University, Hohhot 010070, China; lrs0729@163.com (R.L.); zhanglinmg@aliyun.com (L.Z.); 2Key Laboratory of Animal Genetics, Breeding and Reproduction of Shaanxi Province, College of Animal Science and Technology, Northwest A&F University, Xianyang 712100, China; daixuelei2014@163.com; 3State Key Laboratory of Genetic Resources and Evolution, Kunming Institute of Zoology, Chinese Academy of Sciences, Kunming 650223, China

**Keywords:** whole-genome bisulfite sequencing, CpG, gene promoter region, fat-tailed sheep, *CAMK2D*

## Abstract

**Simple Summary:**

DNA methylation is an important epigenetic modification and plays an important role in the regulation of gene expression. The study of DNA methylation will help to explore the effects of epigenetic modifications, other than DNA sequence variation, on biological phenotypes and physiological functions, promoting the revolution of livestock selection and breeding practices. DairyMeade sheep (thin-tailed) and Mongolian sheep (fat-tailed) have large differences in their dairy and meat production performances, as well as their tail phenotype, thus providing us with good materials for genomic DNA methylation studies. The below results provided a genome-wide DNA methylation landscape of muscle and tail-fat tissues between DairyMeade sheep and Mongolian sheep and a series of differentiated methylation regions (DMRs) in which *CAMK2D* may play a crucial role in fat metabolism and meat quality traits. These results may help us to understand how DNA methylation affects the economic traits of domestic animals like sheep.

**Abstract:**

This study aimed to explore the genome-wide DNA methylation differences between muscle and tail-fat tissues of DairyMeade sheep (thin-tailed, lean carcass) and Mongolian sheep (fat-tailed, fat-deposited carcass). Whole-genome bisulfite sequencing (WGBS) was conducted and the global DNA methylation dynamics were mapped. Generally, CGs had a higher DNA methylation level than CHHs and CHGs, and tail-fat tissues had higher CG methylation levels than muscle tissues. For DNA repeat elements, SINE had the highest methylation level, while Simple had the lowest. When dividing the gene promoter region into small bins (200 bp per bin), the bins near the transcription start site (±200 bp) had the highest CG count per bin but the lowest DNA methylation levels. A series of DMRs were identified in muscle and tail-fat tissues between the two breeds. Among them, the introns of gene *CAMK2D* (calcium/calmodulin-dependent protein kinase II δ) demonstrated significant DNA methylation level differences between the two breeds in both muscle and tail-fat tissues, and it may play a crucial role in fat metabolism and meat quality traits. This study may provide basic datasets and references for further epigenetic modification studies during sheep genetic improvement.

## 1. Introduction

Lamb is popular worldwide for its low cholesterol, low fat, and high protein content. However, in many countries, including China, low growth rates and slaughter rates are commonly seen in indigenous breeds like Mongolian sheep and affect meat production efficiency. The improvement of mutton production efficiency on a large scale is an important issue to be solved urgently. At present, many genetic polymorphisms affecting sheep growth and reproduction have been studied by genome resequencing and whole-genome association analysis [1,2]. However, these widely concerned economic traits still could not be fully explained by polymorphism or quantitative trait loci.

DNA methylation occurs on cytosine residues and is an important epigenetic regulatory mechanism in eukaryotes. It plays a key role in mediating gene expression, genomic imprinting, cell differentiation, embryogenesis, and other biological processes, and determines the phenotypic plasticity of organisms [3]. Whole-genome DNA methylation sequencing has been extensively carried out in many mammals [4,5,6,7]. In terms of litter size, the reproductive efficiency of sheep has an important impact on the economic efficiency of farmers. Reproductive traits usually have low to moderate heritability and do not respond significantly to phenotypic selection. However, studies have shown that ovarian development is affected by DNA methylation [8,9,10]. DNA methylation affects sexual maturation and ovarian maturation in pigs [11]. Similarly, DNA methylation also affects a variety of biological phenotypes. DNA methylation is closely related to changes in gene expression profiles and affects body size [12,13]. In addition, sheep genome-wide association analysis revealed that MBD5 (Methyl-CpG-binding domain protein 5) may be closely related to sheep meat production traits. This gene has a structural domain binding to methylated DNA, suggesting that DNA methylation may be involved in post-weaning development and is a candidate gene for weight gain after weaning [1].

Muscle development and growth were closely related to the proliferation, fusion and differentiation of myoblasts into muscle fibers [14]. These processes are not only affected by genotype, but also by a series of complex epigenetic regulatory mechanisms, including DNA methylation [15,16]. A study based on genome-wide DNA methylation in skeletal muscle of different pig breeds found that differential methylation regions in gene promoters were associated with known obesity-related genes such as *FTO*, *ATP1B1* and *COL8A2* [17]. In normal aging human skeletal muscle tissue, DNA methylation plays a key role in improving proteolytic reactions related to muscle function [18]. A comparative analysis of Japanese Wagyu and Chinese Red Steppes cattle by genome-wide DNA methylation has also indicated that certain DMRs and differentially expressed genes are related to muscle development [19].

Dairy sheep breeds, including East Friesian and DairyMeade, were introduced into China in recent years to establish its own sheep dairy industry. DairyMeade, which originated in New Zealand, is a large-framed sheep breed with good milking performance. Meanwhile, it also grows fast, has a thin tail, good slaughter performance, and a relatively lean carcass. As an indigenous sheep breed that is widely distributed in Inner Mongolia, Mongolian sheep have fat tails, a general slaughter performance, and a fat-deposited carcass. Crossbreeding of DairyMeade and Mongolian sheep may yield a new dual-purpose sheep breed that has both milking and growth performance, while the underlying genetic and epigenetic mechanisms of the trait differences between these two breeds need to be revealed. To explore how DNA methylation affects the muscle growth and fat deposition of these sheep, we collected the muscle and tail-fat tissues from these two sheep breeds as well as their hybrid F_1_ offspring and conducted WGBS. Then the genome-wide DNA methylation landscape at a single-base resolution was drawn, and the differences and differentially methylated genes were identified.

## 2. Materials and Methods

### 2.1. Sampling for Whole-Genome DNA Methylation Sequencing

The sheep used in this study were raised under the same conditions and had free access to feed, water, and access to indoor and outdoor spaces. All animal experiment procedures were performed in accordance with the guidelines approved by the Ethics Committee of Inner Mongolia University. Sampling was carried out according to the “Guidelines on Ethical Treatment of Experimental Animals” established by the Ministry of Science and Technology, China.

A total of 12 biceps femoris muscle samples were collected from 12 male sheep, including 3 three-month-old DairyMeade sheep (DM3M), 3 twelve-month-old DairyMeade sheep (DM12M), 3 twelve-month-old Mongolian sheep (MG12M), and 3 twelve-month-old hybrid F_1_ (F112M, Mongolian sheep x DairyMeade sheep). A total of 10 tail-fat tissues were collected from 10 male sheep, including 4 twelve-month-old Mongolian sheep, 3 twelve-month-old DairyMeade sheep (DM), and 3 twelve-month-old hybrid F_1_ (F1TF). All samples were stored in liquid nitrogen immediately after collection.

### 2.2. DNA Extraction, WGBS Library Preparation and Sequencing

Genomic DNA was extracted from tail-fat and muscle tissues using a DNA extraction kit (Tiangen, Beijing, China). The concentration and quality of DNA were evaluated by agarose gel electrophoresis and the Nano-Drop spectrophotometer. DNA was fragmented by ultrasonication, the fragments were end-repaired, 3′-end-adenylated and ligated with adapters. Agarose gel electrophoresis was used to select fragments of 400–500 bp in length. Finally, the selected fragments were treated with bisulfite and subjected to PCR amplification to construct the sequencing library. The qualified library was subsequently sequenced using an Illumina HiSeq 2500 system with an average genome coverage of 20.78× (Appendix A).

### 2.3. Reads Mapping and DNA Methylation Detection

The WGBS data were analyzed according to the flow chart in Appendix A. The raw reads produced by the Illumina HiSeq 2500 system were filtered for subsequent analysis to ensure the quality of data analysis, including the removal of reads that have adapters and filtration of reads with low-quality bases or more than 10% N content. Then the module bs_seeker2-align.py in BS-Seeker2 software [20] was used to align the clean reads to the sheep reference genome (Oar_rambouillet_v1) with parameter setting --aligner = bowtie2 --bt2-p 8, and indexes were created using the module bs_seeker2-build.py in BS-Seeker2 software with parameter setting --aligner = bowtie2. Finally, the module bs_seeker2-call_methylation.py in BS-Seeker2 software was used to extract information about genomic cytosine sites from the results of the comparison of the clean reads with the reference genome, and thereby acquire the coverage of cytosine site and the number of different types of methylated cytosine (CG as CpG, CHG and CHH).

### 2.4. Annotation of Genomic Regions and Methylation Level Calculation

To evaluate the dynamics of DNA methylation in different genomic regions, a region from 15 kb upstream to 15 kb downstream of each gene (±15 kb) was selected based on the structure and annotation of RefSeq genes, which was then divided into intragenic (or gene body) and intergenic regions. The intragenic region is from the transcription start site (TSS) to the transcription end site (TES) of each gene. The intergenic region is defined as the complementary region of the intragenic region, which consists of 15 kb upstream and 15 kb downstream of each gene. The annotated files of repeat elements of the reference genome were downloaded from NCBI. Four typical types of repeat elements, such as LINE, SINE, LTR, and Simple, and the annotation information of intragenic regions, intergenic regions, exon and intron were extracted to calculate their DNA methylation levels, respectively.

Gene promoters are key regions of DNA methylation to regulate gene transcription. To further explore the dynamics of DNA methylation in promoter regions, the TSS ± 10 kb genome region was first extracted and divided into 100 subregions, and the CG count/amount of each region was calculated to screen the candidate promoter region. Empirically, the TSS ± 2 kb region was defined as a promoter region based on CG count and was further divided into 20 consecutive bins (200 bp for each, termed bin−10…bin10) [21]. The number of CG dinucleotides in each bin was calculated and plotted into a CG frequency histogram. Based on the number of CGs, gene promoters were divided into high CG density (HCG), intermediate CG density (ICG) and low CG density (LCG) groups.

The DNA methylation levels of the gene body and intergenic regions were summed up to determine the global DNA methylation levels of the RefSeq genes. For each genomic region, only CG sites with at least threefold read coverage were considered to calculate the DNA methylation. The average DNA methylation level of all the CGs captured in a region was interpreted as the average DNA methylation level of the region. The DNA methylation of single CG was also limited to the CGs with at least a threefold read coverage. We defined the DNA methylation level of each sample as the arithmetic average of the DNA methylation levels of all the RefSeq genes, whereas the DNA methylation level of each reported stage was defined as the arithmetic average of all the biological replicates. The data was processed with customized Python (3.6.12) scripts. The graphics were drawn with R (3.5.0).

### 2.5. Identification of DMR-Containing Genes

A 600 bp sliding window was used to scan the genome with a step length of 300 bp to calculate the methylation levels in each window to screen the differentially methylated regions (DMRs). Only those windows containing at least 3 CGs with at least threefold read coverage were retained to calculate methylation levels. The methylation level of each window was defined as the arithmetic mean of all CG methylation levels in the window. When the absolute value of the methylation level difference between two samples was greater than or equal to 0.15, the sliding window was considered a differential methylation window. Gene annotation of the reference genome (Oar_rambouillet_v1) was used to annotate DMR-containing genes. The following criteria were used to annotate the DMR-containing genes: (1) Gene fragment contained in DMR; (2) Gene fragment overlapped with DMR; (3) DMR covered by gene fragment. As long as one of these criteria was met.

## 3. Results

### 3.1. Global DNA Methylation Patterns

We first compared the global DNA methylation patterns of muscle and tail-fat tissues. The DNA methylation in different contexts (CG, CHG, and CHH) was calculated. In muscle tissues, 67.82% of CGs had methylation levels greater than 0.5, and 32.18% of CGs had methylation levels lower than 0.5 (Figure 1A). However, DNA methylation levels were relatively lower in both CHGs and CHHs. About 94.37% of CHGs had DNA methylation levels of less than 0.1, and 5.63% of CHGs greater than 0.1. The DNA methylation of CHHs was similar to that of CHGs, with 93.53% of CHHs had DNA methylation levels less than 0.1 and 6.47% of CHGs were greater than 0.1 (Figure 1A). The same pattern was observed in tail-fat tissues (Figure 1B). These results indicate that DNA methylation levels vary in different cytosine contexts throughout the genome. In both tissues, the DNA methylation level of CGs was much higher than that of CHGs and CHHs. Moreover, the global CGs DNA methylation level of tail-fat was slightly higher than muscle.

We then calculated the DNA methylation levels of each sample under different cytosine contexts. The methylation level of CHGs was slightly lower in tail-fat than in muscle (0.044 vs. 0.047), while the methylation level of CHHs was also slightly lower in fat-tail than in muscle (0.050 vs. 0.050). Moreover, the methylation levels in CHHs were slightly higher than in CHGs, both in muscle and tail-fat (Figure 2A).

In the CG context, no significant difference was found in the DNA methylation levels of muscle tissues at two different growth stages. The DNA methylation levels of three-month-old DairyMeade sheep, twelve-month-old DairyMeade sheep, twelve-month-old Mongolian sheep, and F_1_ sheep were 0.668, 0.675, 0.678, and 0.674, respectively. In tail-fat tissues, DNA methylation levels of twelve-month-old DairyMeade sheep, twelve-month-old Mongolian sheep, and F_1_ were 0.716, 0.699, and 0.725, respectively. In general, the methylation levels of CGs for muscle and tail-fat tissues did not differ too much among different groups but the methylation levels of CGs were generally higher in tail-fat tissue than in muscle tissue (Figure 2B).

### 3.2. DNA Methylation Dynamics

DNA methylation in CGs plays an important role in maintaining genomic stability and regulating gene expression. We then focused on CGs and explored the genome-wide dynamics of CG methylation in muscle and tail-fat tissues. The DNA methylation in the gene body and its upstream and downstream 15 kb regions (intergenic region) was calculated and compared. The results showed that, in muscle tissue, the methylation level near TSS was the lowest. The methylation level rapidly increased in the gene body, and reached its peak at TES and then decreased to a relatively stable state at the downstream of TES (Figure 3A). The same patterns were observed in tail-fat tissues and they were not affected by the genetic background of the sheep (Figure 3B).

### 3.3. DNA Methylation of Repeat Elements and Other Genomic Regions

The epigenetic modification of repeat elements plays an important role in gene expression regulation. Based on the genome annotations, we extracted LINE, SINE, LTR, and Simple sequence repeats, as well as the intergenic regions, intragenic region/gene body, exons, and introns to compare their DNA methylation levels in CG contexts. In muscle tissues, the methylation level of SINE was the highest (0.747 on average) and that of Simple was the lowest (0.567 on average). The methylation level of the intergenic region (0.652) was slightly lower than that of the intragenic region (0.697). The methylation levels of exons and introns were quite similar (0.685 and 0.703) (Figure 4A). In tail-fat tissues, the DNA methylation dynamics were similar to muscle tissues, and the methylation level of Mongolian sheep was slightly lower than the others (Figure 4B). In general, the levels of methylation in these repeat elements and genomic regions were slightly higher in tail-fat tissues than in muscle tissues (Figure 4C).

### 3.4. DNA Methylation of Gene Promoter Regions

Gene promoters are commonly defined as the genomic regions flanking TSS that contain conserved DNA sequences for RNA polymerase to recognize and bind, thus starting transcription. Promoter regions are usually rich in CGs and susceptible to epigenetic modifications such as DNA methylation, which affect gene transcription efficiency. To identify candidate promoter regions, genomic regions from 10 kb upstream to 10 kb downstream of each RefSeq gene TSS were extracted and divided into 100 consecutive subregions or bins (200 bp per bin), then the CG count, which means the CG number for each bin, was calculated. The results showed that the bins near TSS (±2 kb) had the highest CG counts while the flank region had a relatively stable low CG count (Appendix A). We then selected the 20 bins in the TSS ± 2 kb region as the promoter region for further analysis. It was clear that the first bin downstream of TSS had the highest CG count (16.84 on average, Figure 5B).

To find out the correlation between CG count and CG methylated level of the promoter region, the gene promoters were divided into three categories according to their CG count in these 20 bins, named HCG (>8 CGs per bin), ICG (4–8 CGs per bin) and LCG (<4 CGs per bin). Then, the dynamic changes of DNA methylation in the gene promoter regions of muscle and tail-fat tissues were compared. In general, the average methylation level of tail-fat tissues (0.357) was similar to muscle tissues (0.346) in the promoter region. However, among these promoters, HCG promoters had the lowest methylation level, and LCG promoters had the highest methylation level in both tissues (Figure 6A). Moreover, HCG and ICG promoters had the lowest methylated bin near TSS (bin-1 and bin1) in both tissues, while the same pattern was not seen in LCG promoters (Figure 6B,C). Since CG counts in the promoter region gradually decreased from TSS to the flanking region, we may conclude that DNA methylation level was negatively correlated with CG density in the promoter region and methylation in the TSS ± 200 bp region may have a core regulatory effect on gene expression.

### 3.5. Analysis of Differentially Methylated Regions

We then scanned the whole genome with a sliding window of 600 bp and a step size of 300 bp and calculated the methylation level for each window for screening DMRs. For tail-fat tissues, twelve-month-old DairyMeade sheep, Mongolian sheep, and F1 individuals were compared in pairs to find DMRs. After strict screening, a total of 34 DMRs were identified among twelve-month-old DairyMeade sheep and Mongolian sheep. Gene annotations for these regions showed that 6 genes were in the list, which were *CAMK2D*, *LOC114110783*, *LOC114110882*, *TSPAN18*, *LOC114110875* and *ADRA1A*. In addition, a total of 20 DMRs were identified and 3 genes were annotated between DairyMeade sheep and F_1_ individuals. Ten DMRs were identified and 2 genes were annotated between Mongolian sheep and F_1_ individuals (Appendix A). For muscle tissues, 14, 12, 12 and 15 DMRs were identified and 4, 4, 1, and 3 genes were annotated in three-month-old vs. twelve-month-old DairyMeade sheep, twelve-month-old DairyMeade sheep vs. twelve-month-old Mongolian sheep, twelve-month-old DairyMeade sheep vs. F_1_ individuals and twelve-month-old Mongolian sheep vs. F_1_ individuals, respectively (Appendix A).

Among these genes, *TSPAN18* is a regulatory factor released by endothelial cell Orai1/Ca^2+^ signal and von Willebrand factor under inflammation stimulation [22]. *ADRA1A* encodes alpha-1 adrenergic receptors (alpha-1-ARs), a member of the G protein-coupled receptor superfamily, play an important role in activating the mitotic response and regulating the growth and proliferation of many cells. GO (Gene Ontology) annotation shows that this gene is related to G protein-coupled receptor activity and α-adrenergic receptor activity. In addition, intranuclear ADRA1A-ADRA1B heterooligomers modulate adrenaline (PE) to stimulate ERK signaling in cardiomyocytes [23]. It is worth noting that the gene *CAMK2D* was detected in both muscle tissue and tail-fat tissue, particularly in tail-fat tissue of twelve-month-old DairyMeade sheep and Mongolian sheep (Figure 7), which encodes calcium/calmodulin-dependent protein kinase II δ and belongs to the serine/threonine-protein kinase family. *CAMK2D* encodes calcium/calmodulin-dependent protein kinase II δ, which is involved in regulating the transport of Ca^2+^, Na^+^ and K^+^ in the sarcoplasmic reticulum and mediates a variety of second messenger effects of Ca^2+^. It may also regulate the transport of sarcoplasmic reticulum Ca^2+^ through phosphorylation and participate in the regulation of skeletal muscle function in response to exercise [24,25].

## 4. Discussion

The effect of epigenetic modifications, including DNA methylation, on the economic traits of domestic animals has not been well studied, especially in grass-feeding livestock. The significant differences in the tail trait, growth rate, body fat rate, and slaughter performance between DairyMeade and Mongolian sheep make them a good model to study the effect of DNA methylation on the above-mentioned traits.

In this study, genome-wide DNA methylation sequencing was performed on muscle and tail-fat tissues of DairyMeade sheep and Mongolian sheep. The genome-wide DNA methylation landscape of muscle and tail-fat tissue of sheep was mapped, and the methylation variations between DairyMeade sheep and Mongolian sheep were compared. Under different cytosine contexts, DNA methylation levels were quite different. DNA methylation was at lower levels in CHH and CHG contexts, which may play an important role in maintaining genome stability [26]. Cytosine in CG context usually has a high methylation level and plays an important role in regulating gene expression, and its methylation level usually differs among different genomic regions [27,28]. The methylation levels of CHH and CHG were both low in the present study, which was consistent with previous studies [15]. We also found that in the sheep genome, the methylation level is region-specific, with a low methylation level near the TSS and a high methylation level near the TES, which was consistent with the results in humans [21]. Although DNA methylation is distributed asymmetrically throughout the genome, the methylation dynamics are completely consistent, and this dynamic pattern is common and unchanged at different developmental stages in mammals, which was also confirmed in our studies that three-month-old and twelve-month-old sheep had similar patterns [29,30]. Furthermore, there are a huge number of repeat elements distributed throughout the genome, which are not inactivated in their natural state and maintain genome configuration and regulate gene expression. DNA methylation patterns and levels of diverse repeat elements are different to some extent, and there is also a certain spatio-temporal specificity [29]. Our results indicated that among the four repeat elements, Simple had the lowest methylation level.

The CG count and DNA methylation level of the promoter region have a greater impact on chromatin accessibility. Low DNA methylation levels in the promoter region are more likely to establish chromatin accessibility, while high DNA methylation levels are the opposite [31]. Combined with our previous results, the GC count and DNA methylation level at promoter regions showed similar variations and distribution patterns between sheep and humans, and DNA methylation flanking the TSS (±200 bp) may be closely related to the regulation of gene expression [21].

Through strict screening, the region located at 15,111,900–15,112,500 bp on chromosome 6 showed significant methylation differences in tail-fat tissue between twelve-month-old DairyMeade sheep and Mongolian sheep. This region is located in the intron of the *CAMK2D* gene (Chr6, 14,727,325–15,282,831 bp). It was reported that alternative splicing of this gene leads to multiple transcriptional variants encoding different protein isoforms and participating in multiple physiological activities [24]. In addition, the Ca^2+^ signaling pathway is involved in regulating the biological clock, intestinal microbial activity, and neuron excitability, thus regulating mammalian food intake, energy metabolism, and adipocyte differentiation [32]. A previous study also suggested that the *CAMK2D* gene is closely related to the fat accumulation of Landrace pigs [33]. *CAMK2D* also contains the gene *LOC114115266* (Chr6, 14,953,544–14,961,128 bp, upstream of the DMR), which encodes ankyrin for connection and integration. We speculate that the differential methylation level in tail-fat tissue may play a role in the selective splicing of *CAMK2D* to produce multiple transcripts, thereby affecting its transcriptional changes and expression differences. Moreover, another region of *CAMK2D* (15,114,600–15,115,200 bp) also showed differential methylation levels in muscle tissue from twelve-month-old DairyMeade sheep and Mongolian sheep, and this DMR may also affect the selective splicing to produce different protein subtypes, thus affecting the muscle development in DairyMeade sheep and Mongolian sheep.

## 5. Conclusions

CG plays a major role in sheep’s genome methylation. Tail-fat tissues had a higher methylation level than muscle tissues. The DNA sequence flanking the TSS (±2 kb) is the candidate promoter region of the sheep gene, and the DNA methylation level was negatively correlated with CG density in the promoter region in those genes which have high CG counts in the candidate promoter region. In those DMRs from tail-fat and muscle tissues between DairyMeade and Mongolian sheep, *CAMK2D* is a promising candidate gene which may regulate the fat and muscle development of sheep through mRNA alternative splicing.

## Figures and Tables

**Figure 1 animals-12-01399-f001:**
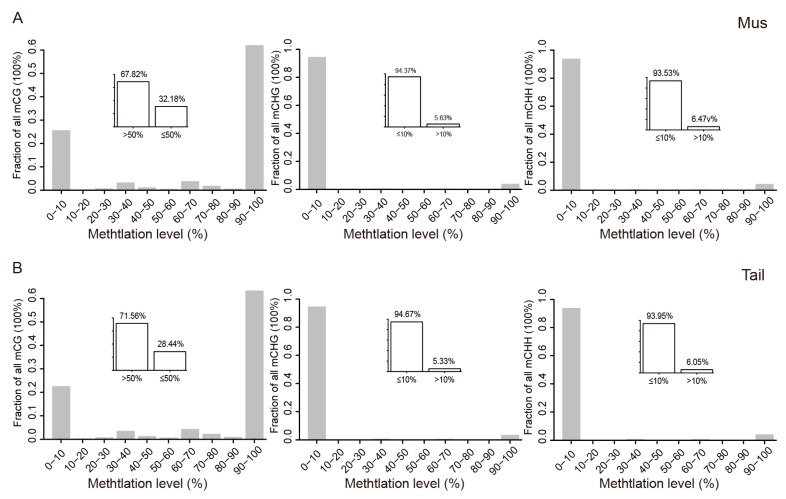
Global DNA methylation patterns. (**A**) Muscle tissue and (**B**) tail-fat tissue of DairyMeade sheep and Mongolian sheep, distribution of DNA methylation under CG, CHH and CHG contexts, only cytosines covered at least 3 times were considered.

**Figure 2 animals-12-01399-f002:**
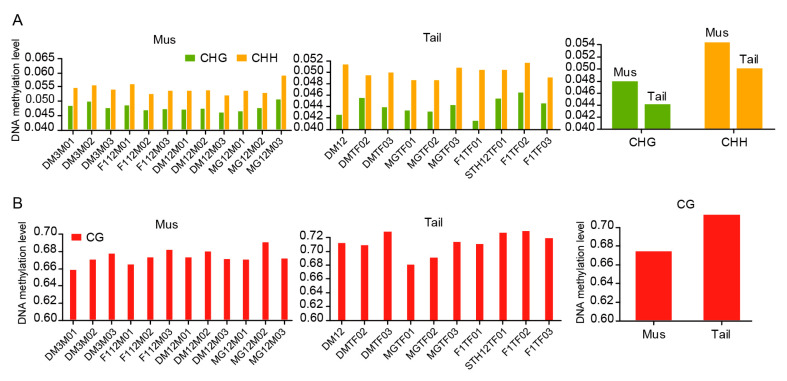
DNA methylation patterns under different cytosine contexts. (**A**) DNA methylation levels of CHG and CHH contexts in tail-fat tissue and muscle tissue, (**B**) DNA methylation level of CG context.

**Figure 3 animals-12-01399-f003:**
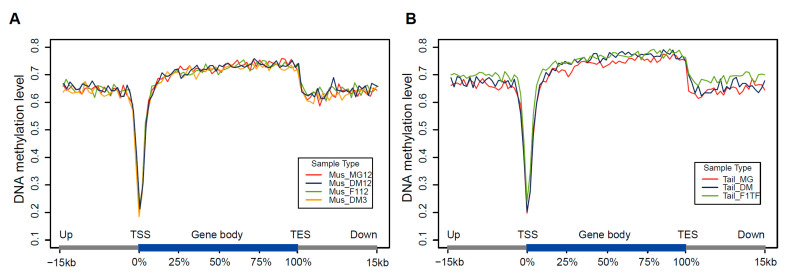
DNA methylation dynamics of gene body and flank regions. DNA methylation dynamics of (**A**) muscle tissue and (**B**) tail-fat tissue, averaged DNA methylation levels along the gene body and 15 kb upstream of TSS and 15 kb downstream of TES of all annotated RefSeq genes.

**Figure 4 animals-12-01399-f004:**
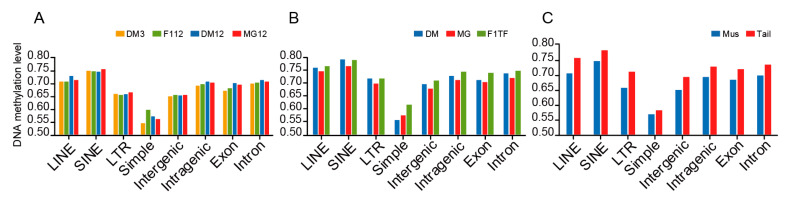
DNA methylation dynamics of repeat elements, intergenic regions, and intragenic regions. DNA methylation dynamics of (**A**) muscle tissue, and (**B**) tail-fat tissue. (**C**) Comparison of DNA methylation levels between muscle tissue and tail-fat tissue.

**Figure 5 animals-12-01399-f005:**
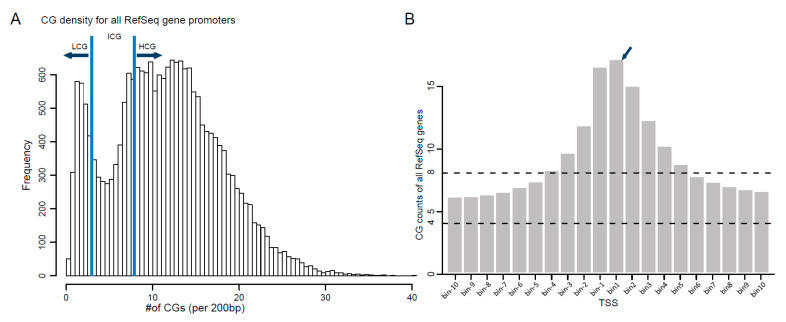
The CG density and distribution of the 20 consecutive bins in promoter regions. (**A**) Classification of the CG density for all RefSeq gene promoters. The promoters were separated into three classes: (1) HCG, more than 8 CGs per 200 bp; (2) LCG, less than 4 CGs per 200 bp; (3) ICG, 4–8 CGs per 200 bp. (**B**) Distribution of CG counts at the 20 bins of all RefSeq genes. The number of CGs at bin1 was the highest.

**Figure 6 animals-12-01399-f006:**
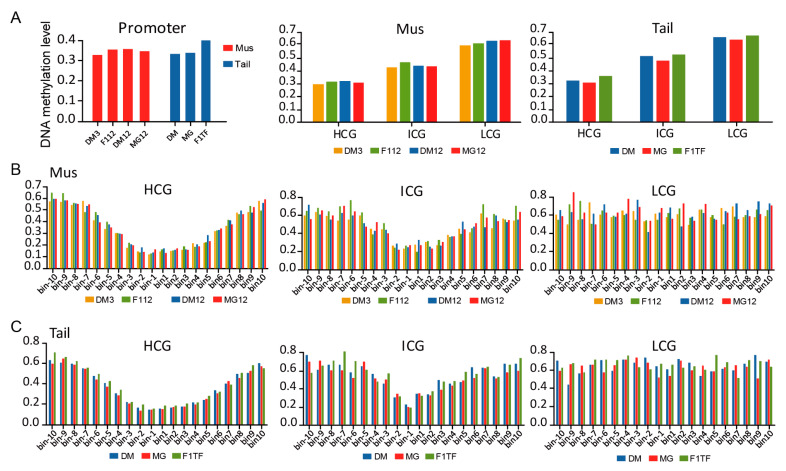
DNA methylation patterns of the promoter region for muscle tissue and tail-fat tissue. (**A**) Global DNA methylation level of prompter region (left), and DNA methylation of HCG, ICG and LCG-types promoter for muscle tissue (middle) and tail-fat tissue (right). (**B**) DNA methylation patterns of muscle tissue and (**C**) tail-fat tissue at the consecutive 20 bins of the promoter region.

**Figure 7 animals-12-01399-f007:**
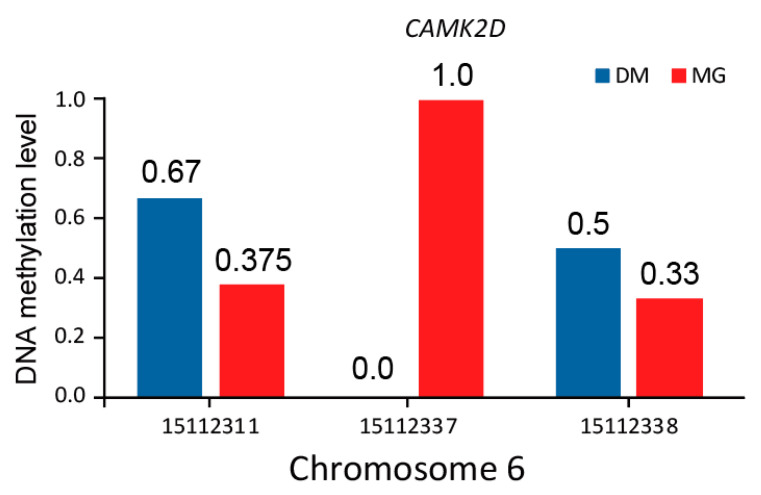
DNA methylation level of differentially methylated sites within *CAMK2D,* all the methylated cytosine were covered more than three times.

## Data Availability

The whole-genome bisulfite sequencing datasets generated in this study were submitted to the National Center for Biotechnology Information (NCBI) Sequence Read Archive with the accession code PRJNA822017. The additional datasets supporting the conclusions in this paper can be found in the Appendix A.

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
