# Peer review of "Genome-Wide DNA Methylation Patterns of Muscle and Tail-Fat in DairyMeade Sheep and Mongolian Sheep"

_animals, 2022, doi:10.3390/ani12111399_

Round 1
Reviewer 1 Report
The authors present an interesting paper on genome-wide DNA methylation patterns of muscle and tail-fat in Chinese sheep breeds.
Although the subject is attractive, a validation of the identify DMGs is missing. In my opinion, this limitation should be addressed, and this make the paper not acceptable in the present form.
Title
Please replace “Gnome-wide” with “Genome-wide”.
Abstract
The abstract should be a total of 200 words maximum.
Line 30. Are two or three the sheep strains studied?
Keywords
It is better to avoid words already included in the title
Introduction
Line 71. Authors must explain acronyms the first time that they use them in the text (DMRs).
Materials and Methods
The authors must explain the method used for DMGs annotation.
In addition, quantitative reverse transcriptase PCR (qRT-PCR) should be used to validate the DMGs from the sequencing results by detecting the expression level of the mRNAs. Can the authors explain why they did not validate the DMGs reported in the manuscript?
Line 109. Maybe the genomic DNA was fragmented before to perform bisulfite conversion. Please clarify
Line 117. Please provide information about the used Illumina platform.
Results
Lines 161-169. Please move this paragraph to material and methods.
Figures 2. Please explain the abbreviations used in the X axis.
References
References must be numbered in order of appearance in the text.
Reviewer 2 Report
Simple summary is missing
I liked the abstract but it exceeded the number of words accepted by the journal. Please shorten. Furthermore, I missed numerical results supporting your claims in the abstract.
Keywords are inappropriate as all of them are already used in the title. Search engines already look for this, hence repeating means you miss a chance to be found and cited.,
Introduction
Citations need to be added to support each different idea. Why do citations start on number 31?
Introduction was a little bit unorganized and hard to follow, although I liked the content. Please try to work a little bit more on it. In the end You do not have a clear statement of which the objectives of the paper are or which the hypotheses to be tested.
M&M
Sample has not been clearly defined. Which are the ages and sexes. How many animals?. Although some of the information is present, it is still unorganized and hard to infer straightforwardly.
Genetic analyses describe the protocol and not much need to be added. They look right to me. Graphs are really nice.
Was any statistical analysis run. If it was it is not reported, nor the software used for the aforementioned analyses and graphs were made using one for sure. Please add.
Discussion
First paragraph is unnecessary and repeated in other sections. Please remove it.
Discussion resembles introduction. Although content is nice and well managed, the organization of the paper still needs a little bit of effort by the authors.
Conclussion section is missing.
Round 2
Reviewer 1 Report
Thanks for your effort in improving the last version of your manuscript.
In my opinion, the validation of DMGs remanins the main concern of this manuscript. I hope in the near future the authors will be able to improve this analytical aspect.
Author Response
Reviewer 1:
Thanks for your effort in improving the last version of your manuscript.
Answer: Thanks for the suggestion. We are also eager to conduct the validation experiment as soon as possible after COVID 19 restriction.
Reviewer 2 Report
The authords did a great job in attending my requests. However, two additional points must be considered.
First, even if R and Python were used (versions, packages, libraries, etc.), the statistical procedure needs to be presented. If own scripts were used, this must be provided as a supplementary file, as otherwise the study is not replicable.
Second, conclussions are not conclussions but a summary of results, what you take home from this results (what these results suggest) is what you need to add here.
Author Response
Reviewer 2:
The authords did a great job in attending my requests. However, two additional points must be considered.
Answer: Thanks.
First, even if R and Python were used (versions, packages, libraries, etc.), the statistical procedure needs to be presented. If own scripts were used, this must be provided as a supplementary file, as otherwise the study is not replicable.
Answer: Thanks for the valuable advice. The versions of Python and R were added in the M&M section. We have 3 replicates for each sample group. Since the data yielded from methylation analysis (methylated or not) didn’t fit the Poisson distribution (not like wet experiments), so we only made the arithmetic mean but not conduct statistical analysis.
Second, conclussions are not conclussions but a summary of results, what you take home from this results (what these results suggest) is what you need to add here.
Answer: Thanks for the suggestion. We have rewritten the conclusions.